# Antimicrobial Resistance Profiles and Co-Existence of Multiple Antimicrobial Resistance Genes in *mcr*-Harbouring Colistin-Resistant *Enterobacteriaceae* Isolates Recovered from Poultry and Poultry Meats in Malaysia

**DOI:** 10.3390/antibiotics12061060

**Published:** 2023-06-15

**Authors:** Md. Rezaul Karim, Zunita Zakaria, Latiffah Hassan, Nik Mohd Faiz, Nur Indah Ahmad

**Affiliations:** 1Department of Veterinary Pathology & Microbiology, Faculty of Veterinary Medicine, Universiti Putra Malaysia (UPM), Serdang 43400, Selangor, Malaysia; reza@blri.gov.bd (M.R.K.); nurindah@upm.edu.my (N.I.A.); 2Bangladesh Livestock Research Institute, Savar, Dhaka 1341, Bangladesh; 3Institute of Bioscience, Universiti Putra Malaysia (UPM), Serdang 43400, Selangor, Malaysia; 4Department of Veterinary Laboratory Diagnostics, Faculty of Veterinary Medicine, Universiti Putra Malaysia (UPM), Serdang 43400, Selangor, Malaysia; latiffah@upm.edu.my; 5Department of Veterinary Clinical Studies, Faculty of Veterinary Medicine, Universiti Putra Malaysia (UPM), Serdang 43400, Selangor, Malaysia; nikmdfaiz@upm.edu.my

**Keywords:** colistin resistance, *mcr*, co-existence, *Enterobacteriaceae*, poultry, Malaysia

## Abstract

The co-existence of the colistin resistance (*mcr*) gene with multiple drug-resistance genes has raised concerns about the possibility of the development of pan-drug-resistant bacteria that will complicate treatment. This study aimed to investigate the antibiotic resistance profiles and co-existence of antibiotic resistance genes among the colistin-resistant *Enterobacteriaceae* isolates recovered from poultry and poultry meats. The antibiotic susceptibility to various classes of antibiotics was performed using the Kirby-Bauer disk diffusion method and selected antimicrobial resistance genes were detected using PCR in a total of 54 colistin-resistant *Enterobacteriaceae* isolates including *Escherichia coli* (*E*. *coli*) (n = 32), *Salmonella* spp. (n = 16) and *Klebsiella pneumoniae* (*K*. *pneumoniae*) (n = 6) isolates. Most of the isolates had multi-drug resistance (MDR), with antibiotic resistance against up to seven classes of antibiotics. All *mcr*-harbouring, colistin-resistant *Enterobacteriaceae* isolates showed this MDR (100%) phenotype. The *mcr-1* harbouring *E*. *coli* isolates were co-harbouring multiple antibiotic resistance genes. The seven most commonly identified resistance genes (*^bla^TEM*, *tetA*, *floR*, *aac-3-IV*, *aadA1*, *fosA*, *aac*(*6_*)*-lb*) were detected in an *mcr-1*-harbouring *E*. *coli* isolate recovered from a cloacal swab. The *mcr-5* harbouring *Salmonella* spp. isolate recovered from poultry meats was positive for *^bla^TEM*, *tetA*, *floR*, *aac-3-IV*, *fosA* and *aac*(*6_*)*-lb* genes. In conclusion, the colistin-resistant *Enterobacteriaceae* with *mcr* genes co-existing multiple clinically important antimicrobial resistance genes in poultry and poultry meats may cause potential future threats to infection treatment choices in humans and animals.

## 1. Introduction

Antibiotics are prescribed to treat infections caused by bacterial strains, including Enterobacteriaceae, but the prevalence of antibiotic resistance has reached epidemic proportions in humans and animals, acknowledged as a rising severe threat to public health and food safety globally. Human medications used as growth stimulants or animal preventive agents have been blamed for exacerbating the global dilemma of antibiotic resistance [1].

The spread of drug-resistant *Enterobacteriaceae* infections was also facilitated by improper antibiotic usage in humans and animals as well as greater accessibility worldwide [2]. Multidrug-resistant (MDR) (resistant to three or more different antimicrobial classes), organisms, including ESBL, are commonly known as “superbugs” that severely limit potential treatments and are linked to higher mortality, morbidity, and financial consequences [3]. Multidrug-resistant (MDR) *Escherichia coli*, *Salmonella* spp. and *Klebsiella pneumoniae* are significant members of *Enterobacteriaceae* that are capable of acquiring resistance to various types of antibiotics including aminoglycosides, carbapenems, fluoroquinolones and cephalosporins [4].

Colistin is a positively charged polypeptide antibiotic and is used significantly to treat Gram-negative bacterial infections [5]. Nevertheless, due to their neuro- and nephrotoxicity, as well as the availability of comparatively “safer” medications such as beta-lactams, they were no longer prescribed in the 1980s [6]. Colistin was reintroduced in the 1990s to combat the uncontrollable spread of XDR (extensively drug-resistant) and MDR bacteria, including carbapenem-resistant bacterial strains, despite their continued harmful effects [7]. Colistin was recognised as one of the Human Highest Priority Critically Important Antimicrobials (HPCIA) by the World Health Organization (WHO) since 2017 [8,9]. Because of the lack of new, potent, and safer antibiotics, colistin plays a pivotal role in modern healthcare as a last-line antibiotic for treating severe infections [10]. Like all other antibiotics, bacteria have developed resistance to colistin, as evidenced by several recent studies indicating the advent and global dissemination of colistin resistance [8,11,12].

The plasmid-mediated colistin resistance *mcr* gene was first reported in China in November 2015 in numerous bacterial strains, including *E*. *coli*, *Salmonella* spp. and *Klebsiella pneumoniae*, isolated from aquatic environments, humans and animals [11,12,13]. Until 2015, colistin resistance was thought to be caused by alterations of chromosomally mediated-genes, which disseminate vertically and slowly [14]. Nevertheless, the acquisition of various colistin resistance (*mcr*) genes, including *mcr-1* to *mcr-10* encoded on plasmids, has sped up the dissemination of colistin resistance.

*Salmonella* isolates in chicken carcasses in Peninsular Malaysia have shown sensitivity to cephalosporin, but 5% isolates have shown multidrug resistance in which the highest resistance was observed against streptomycin (66.6%), followed by tetracycline (44.3%) and trimethoprim-sulfamethoxazole (16.6%) [15]. The occurrence of ESBL- *E*. *coli* was 48.8 % in poultry meat in Malaysia [16]. Retail poultry meat has been identified in Malaysia as a common reservoir of multi-drug resistant Salmonella, accounting for 6.7% of *S*. *enteritidis* [17]. In another report, 27 (57.45%) of the Salmonella isolates from Malaysian chicken and chicken meat were found to be resistant to one or more antibiotics tested [18]. Genotypic determinants for resistance, such as *floR, cmlA, tetA, tetB, tetG, temB*, *blaPSE-1, sul1, sul2, qnrA, qnrS, strA* and *aadA,* have been identified among MDR Salmonella strains [19]. In Malaysia, colistin resistance with the *mcr-1* gene was first noted in *E*. *coli* isolated from chicken [20,21]. The co-resistance to β-lactamases and colistin antibiotics were also observed in *K*. *pneumoniae* isolates in swine [22]. The co-existence of colistin resistance (*mcr*) gene with genes encoding resistance to multiple antibiotics, such as carbapenem, extended-spectrum β-lactam, tetracycline, sulfamethoxazole, ciprofloxacin and trimethoprim, raises concerns about the possibility of development of pan-drug-resistant bacteria [23,24]. Hence, the purpose of this research was to determine the antimicrobial resistance profile and co-existence of *mcr* genes with other multiple antibiotic resistance genes among the colistin-resistant *Enterobacteriaceae* isolates recovered from poultry and poultry meats.

## 2. Results

### 2.1. Colistin Resistance in the Isolates

In broth microdilution tests, all the 54 isolates were classified as colistin resistant with MIC values ranging from 4–128 µg/mL colistin. In molecular detection for colistin resistance, *mcr* gene was observed in seven isolates, including six *E*. *coli* isolates with *mcr-1* gene and 1 *Salmonella* spp isolate with *mcr-5* gene.

### 2.2. Phenotypic Detection of Antimicrobial Resistance

#### 2.2.1. Rate of Antimicrobial Resistance to *Enterobacteriaceae*

Antimicrobial resistance (%) rates were grouped and labelled based on the following rate ranges: >70%, >50 to 70%, >20 to 50%, >10 to 20%, >1 to 10%, 0.1 to 1% and <0.1% as extremely high, very high, high, moderate, low, very low and rare, respectively, according to Papadopoulos et al. (2021) and Adebowale et al. (2022). According to this categorization scheme, it was found that the isolated colistin-resistant *E*. *coli* rates were: extremely high in resistance to tetracycline (93.8%); very high in resistance to streptomycin (59.4%) and nalidixic acid (56.3%); and high in rates of resistance to ciprofloxacin and gentamycin (46.9%), tobramycin (43.8%), chloramphenicol (40.6%), cefotaxime (34.4%), norfloxacin (31.3%), ceftriaxone (28.1%) and fosfomycin (25%).

In the case of *Salmonella* spp., very high rates of resistance were found against tetracycline (68.8%), high rates of resistance to chloramphenicol (37.5%), nalidixic acid (31.3%), streptomycin (25%) and ciprofloxacin (25%) and gentamycin (25%). In addition, a moderate resistance rate was observed for tobramycin (12.5%), and a low resistance rate was found for cefotaxime and ceftriaxone (6.3%).

As for *Klebsiella pneumoniae* isolates, extremely high resistance rates were noted for ciprofloxacin (83.3%), very high rates of resistance were observed for tetracycline (66.7%) and chloramphenicol (50%), high resistance rates were detected against streptomycin and nalidixic acid (33.3%). Figure 1 presents an overview of the antimicrobial resistance profiles among the 54 isolates.

Colistin-resistant *E*. *coli* isolates originating from litter samples showed 100% (2, 2/2) resistance to streptomycin, ciprofloxacin, tetracycline, norfloxacin, nalidixic acid and chloramphenicol. The isolates originated from litter samples showed the highest resistance to these antibiotics compared with isolates recovered from cloacal swabs and meat samples. On the other hand, there was a low rate of resistance to fosfomycin (1, 1/7, 14.3%) for the *E*. *coli* isolated from the cloacal swab. (Figure 2).

Colistin-resistant *Salmonella* spp. isolates obtained from cloacal swabs showed 100% (3, 3/3) resistance to tetracycline and nalidixic acid, and isolates recovered from litter samples showed 100% (2, 2/2) resistance to ciprofloxacin and tetracycline. In contrast, the highest rate from meat samples, 54.5% (n = 6, 6/11) of isolates, was found to be resistant to tetracycline (Figure 2).

Colistin-resistant *K*. *pneumoniae* isolates obtained from cloacal swabs and litter samples showed 100% resistance to tetracycline and chloramphenicol, respectively. Ciprofloxacin resistance was observed in 100% isolates originating from litter samples and meat samples (Figure 2).

#### 2.2.2. Antimicrobial Resistance Profile

Multidrug resistance (MDR) was found in the majority of *Enterobacteriaceae* isolates (85.19%, 46/54). This total was further summarized by species groups. Among the colistin-resistant *E*. *coli* isolates, 94% (n = 30) of isolates were MDR, with five different patterns of resistance against three (3) to seven (7) classes of antibiotics. *E*. *coli* isolates had the largest number with resistance to seven classes of antibiotics, 16% (n = 5) (Figure 3). Among *Salmonella* spp. isolates, 68.5% (n = 11) were MDR, with four patterns. The highest proportion of these MDR isolates, 31% (n = 5), were found to be resistant to four classes of antibiotics, followed by 25% (n = 4) of isolates with resistance to three classes of antibiotics. On the other hand, 6% (n = 1) *Salmonella* spp. isolates had resistance to six classes of antibiotics, which was the highest number of antibiotic classes for this group (Figure 3). In *K*. *pneumoniae*, 83% (n = 5) of isolates were MDR, with only two patterns, and 67% (n = 4) of isolates were found to be resistant to four classes of antibiotics (Figure 3). All seven *mcr*-habouring colistin-resistant *Enterobacteriaceae*, including *E*. *coli* (n = 6) and *Salmonella* spp. (n = 1) were of the MDR phenotype (Table 1).

#### 2.2.3. Multiple Antibiotic Resistance Index (MARI)

A total of 54 colistin-resistant isolates showed various antibiotic resistance patterns in the current study. Among these isolates, 85.19% (46/54) had MARI greater than 0.2, from which 38.89% (21/54) of isolates showed MARI of more than 0.4. Out of 32 *E*. *coli*, 93.75% (n = 30) and 62.5% (n = 20) of *E*. *coli* isolates showed MARI more than 0.2 and 0.4, respectively. The highest MARI value of 0.79 was observed in two *E*. *coli* isolates which were recovered from cloacal swabs and chicken meat samples (Appendix A). MARI of 0.29 was found in 21.9% of isolates, followed by MARI of 0.43 in 18.8% of isolates, and the highest MARI of 0.79 was determined in 6% of isolates (Appendix A). In total, 68.75% (n = 11) and 6.25% (n = 1) of *Salmonella* spp. isolates had MARI values of more than 0.2 and 0.4, respectively (Appendix A). MARI of 0.21 was observed in 25% of isolates, and the highest MARI of 0.71 was noted in 6.25% of isolates (Appendix A). However, 83.33% (n = 5) of *K*. *pneumoniae* isolates showed MARI of >0.2. None of the *K*. *pneumoniae* isolates had MARI higher than 0.4 (Appendix A). The majority of the *K*. *pneumoniae* isolates had MARI of 0.29 and 0.36 (Appendix A).

### 2.3. Other Antibiotic Resistance Genes in Colistin-Resistant Enterobacteriaceae

Among colistin-resistant *Enterobacteriaceae* isolates, 51.9% (n = 28) were positive for the *tetA* gene, followed by ^bla^*TEM* (35.2%, n = 19) and *floR* (35.2%, n = 19) genes. Whereas only 1.9% (n = 1) isolate showed positive for the *catA1* gene (Figure 4). The presence of *tetA*, *floR*, *aadA1*, *fosA*, and *aac*(*6_*)*-lb* genes were observed in *E*. *coli*, *Salmonella* spp., and *K*. *pneumoniae* isolates. Two genes, ^bla^*TEM* and *aac-3-IV*, were found both in *E*. *coli* and *Salmonella* spp. isolates. The ^bla^*SHV* gene was detected only in *K*. *pneumoniae* isolates (66.67%), and the occurrence of *catA1* genes was noted only in *E*. *coli* isolates. The resistance gene that was most common among all colistin-resistant *Enterobacteriaceae* was the *tetA* gene, detected in 53.13% of *E*. *coli*, 50% of *Salmonella* spp., and 50% of *K*. *pneumoniae* isolates (Figure 5).

The β-lactamase genes and other antimicrobial resistance genes were observed in colistin-resistant *Enterobacteriaceae*. Specifically, 33.33% had the ^bla^*SHV* gene, 83.3% had the *tetA* gene, 83.3% had the *floR* gene and 66.7% had the *aadA1* gene, and these were found to be significantly more common in *Enterobacteriaceae* isolated from litter samples than from other sources (Table 2, *p* < 0.05). Out of 54 colistin-resistant *Enterobacteriaceae* isolates, 29 isolates were positive for at least one resistance gene, in which 18 different patterns of resistance were observed (Table 3).

All colistin-resistant *E*. *coli* isolated from litter samples were found to harbour *tetA* and *floR* genes. This contrasted with isolates from cloacal swabs and meat samples, although no statistically significant difference was found (*p* > 0.05). The *aadA1* gene was found in 100% of colistin-resistant *E*. *coli* isolated in litter samples. This frequency was significantly higher than that among *E*. *coli* obtained from cloacal swabs and meat samples (Appendix A, *p* < 0.05). In contrast, the *catA1* gene was confirmed in 4.3% of colistin-resistant *E*. *coli* isolated from meat samples. Out of 32 colistin-resistant *E*. *coli* isolates, 17 isolates were detected with at least one resistance gene, in which 11 different patterns of genotypic resistance were observed. Resistance to seven genes (*^bla^TEM*, *tetA*, *floR*, *aac-3-IV*, *aadA1*, *fosA*, *aac*(*6_*)*-lb*) was detected in one isolate (Table 3).

All colistin-resistant *Salmonella* spp. isolated from the cloacal swab harboured the *tetA* gene, and no other resistance genes were detected. The *tetA* gene was also detected from *Salmonella* spp. obtained from two other sources. The *^bla^TEM*, *floR* and *fosA* genes were detected from *Salmonella* spp. of litter and meat samples (Appendix A). Out of 16 colistin-resistant *Salmonella* spp. isolates, eight isolates were positive for at least one resistance gene, in which four different patterns of resistance were observed. The highest number of resistance genes (*^bla^TEM*, *tetA*, *floR*, *aac-3-IV*, *fosA*, *aac*(*6_*)*-lb*), six, were detected in one isolate, followed by five genes in one isolate, four genes in three isolates and one gene in 3 isolates (Table 3).

Colistin-resistant *K*. *pneumoniae* isolates originated from three different sources were found to be positive for *^bla^SHV* and *fosA* genes, whereas *tetA*, *floR*, *aadA1*, and *aac*(*6_*)*-lb* genes were detected from the isolates obtained from cloacal swab and litter samples. No other genes were observed in *K*. *pneumoniae* isolates (Appendix A). Out of 6 colistin-resistant *K*. *pneumoniae* isolates, four isolates were positive for at least two resistance genes, in which, three different patterns of resistance were observed. Six resistance genes (*^bla^SHV*, *tetA*, *floR*, *aadA1*, *fosA*, *aac*(*6_*)*-lb*) were detected in two isolates, followed by three genes in one isolate and two genes in one isolate (Table 3).

The *mcr*-habouring *Enterobacteriaceae* isolates also possessed multiple antibiotic resistance genes (Table 4).

## 3. Discussion

Antimicrobial resistance (AMR) is a significant worldwide public health hazard, according to the World Health Organization, and it is predicted that AMR-related mortality could reach 10 million deaths by 2050 and cost 100 trillion USD [25]. In the population, colistin resistance, *mcr* genes may continue to be maintained due to the co-existence of resistance genes for the other antimicrobials [26,27]. Resistance to multiple classes of antimicrobials were tested in this investigation of the frequency of multidrug resistance with colistin. This current study revealed the varying degree of resistance to different classes of antibiotics. It is widely acknowledged that the poultry sector is a significant environment for the global spread of AMR due to the indiscriminate use of antibiotics in poultry agriculture [25]. We noted high resistance rates to frequently prescribed antibiotics and significant prevalence of MDR against 3 to 11 antibiotics. We also observed various patterns of AMR in colistin-resistant *Enterobacteriaceae* isolates recovered from chicken meats, cloacal swabs, and litter samples. Colistin-resistant *E*. *coli* isolates were found to be highly resistant to tetracycline, followed by streptomycin, nalidixic acids, ciprofloxacin, gentamicin, tobramycin, chloramphenicol, and cefotaxime. A recent study in Bangladesh found that 100% of colistin-resistant *E*. *coli* isolates were resistant to tetracycline, followed by erythromycin (92%), nalidixic acid (77%), gentamycin (62%), ciprofloxacin (46%), chloramphenicol (15%) and cefotaxime (8%), and all *mcr-1* bearing isolates were sensitive to imipenem and fosfomycin [28]. Similarly, all *mcr-1*-habouring *E*. *coli* isolates were sensitive to imipenem and meropenem, but three out of seven isolates were resistant to fosfomycin in our study. In contrast, *mcr-1*-bearing *E*. *coli* isolates recovered from poultry in Nepal were reported resistant to imipenem (32.9%) and meropenem (2.6%) [8]. Modifying the mechanism of carbapenemase production could have led to the development of bacterial strains’ resistance to extended-spectrum beta-lactamases (ESBLs). MDR was found in almost all colistin-resistant *E*. *coli* isolates, showing various patterns of MDR against three to seven classes of antibiotics. Sixteen percent of isolates were observed to be resistant to the highest number of antibiotic classes (seven classes, including 11 antibiotics). A study from Nepal showed that 80% of *E*. *coli* recovered from poultry were MDR [8]. The presence of colistin-resistant MDR *E*. *coli* is threatening in the areas where fatality due to infectious diseases is frequent.

All colistin-resistant *Salmonella* spp. isolates were sensitive to meropenem, imipenem and fosfomycin and highly resistant to tetracycline, followed by chloramphenicol and nalidixic acids, and very few were resistant to cefotaxime and ceftriaxone. According to a study on *mcr*-positive isolates, they remain sensitive to many other antibiotics [29]. In another study, *mcr-1* positive *Salmonella typhimurium* strains showed that all (n = 3) were resistant to cefotaxime and cefepime and sensitive to meropenem, imipenem and fosfomycin [30]. Most of the colistin-resistant *Salmonella* spp. isolates demonstrated MDR, phenotypically, which is corroborated with a recent study, in which almost all (92.3%, 48/52) *mcr*-positive isolates were MDR [31].

The isolated colistin-resistant *K*. *pneumoniae* strains were phenotypically resistant to five classes of antibiotics, and high rates of resistance were observed to ciprofloxacin, tetracycline and chloramphenicol. Colistin-resistant *Klebsiella pneumoniae* isolates in Taiwan were resistant to 17 antimicrobials, and 67.3% of isolates were resistant to ciprofloxacin [32]. In contrast, most *K*. *pneumoniae* isolates in Pakistan were documented to be resistant to colistin (70%), but all showed resistance to tetracycline [33].

Among the studied colistin-resistant isolates, 18 various antibiotic resistance patterns were identified. Among these isolates, 85.19% (46/54) had a MAR index greater than 0.2, from which 38.89% (21/54) showed MAR index >0.4. Species-wise, 93.75% of *E*. *coli*, 68.57% of *Salmonella* spp., and 83.33% of *K*. *pneumoniae* isolates were found to have MARI values of >0.2. A MAR index of >0.2 denotes the overuse or abuse the antimicrobials in humans and animals, and a MAR index of >0.4 suggests contamination from feces as a source. On the other hand, a MAR index ≤0.2 indicates the infrequent or no use of antibiotics, and a MAR index ≤0.4 implies contamination with non-human feces [34]. Our findings are consistent with results in poultry carcasses in Egypt, in which 86.8% of isolates were reported with MAR index value >0.2 [35], and more than 94% of *E*. *coli* isolates were documented with MARI value of >0.2 in South Africa [36]. It might be dangerous to both public health and the poultry sector.

*Escherichia coli* is a significant source of the resistance genes that have been linked to both human and animal therapeutic failure [37]. Most of the colistin-resistant *E*. *coli* isolates harboured the *tetA* gene followed by *^bla^TEM*, *floR*, *aadA1*, *aac-3-IV*, *aac*(*6_*)*-lb*. A few isolates were positive for *catA1* and *fosA* genes. None of the *E*. *coli* isolates possessed the *^bla^SHV* gene. Two out of 13 colistin-resistant *E*. *coli* isolates in Bangladesh were documented to have both the *mcr-1* and *^bla^TEM* genes [28]. The co-existence of the plasmid-borne ESBL and *mcr-1* genes may help spread of resistance to colistin [38]. It may help superbugs develop and spread, making them resistant to all antibiotics currently available on the market. Furthermore, we observed multiple AMR genes in 15 of the 32 colistin-resistant *E*. *coli* isolates, which may indicate the co-transfer of various AMR genes. Beta-lactamase and fluoroquinolone resistance genes were found together and were prominent in our isolates. Beta-lactams and fluoroquinolones are both significant and frequently prescribed antimicrobials in clinical settings. Public health could be harmed by the co-transfer of these two categories of genes [39]. While *mcr-4*-habouring isolates were reported to co-exist with the *catA1* gene, which encodes chloramphenicol resistance, in Spain and Belgium [40], in our study, the *catA1* gene was found in only one *mcr-1*-habouring colistin-resistant *E*. *coli* isolate. According to our research, *E*. *coli* from chicken meat may be a source of resistance genes that could spread to other common bacteria. Despite not exhibiting the phenotype, few isolates carried resistance genes, which may have been caused by the genes being silenced or under transcriptional regulatory control [39]. To better understand the mechanism, more research is necessary.

Colistin-resistant *Salmonella* spp. isolates were found to be positive for *tetA*, *TEM*, *floR*, *aac-3-IV*, and *aac*(*6_*)*-lb* genes, which were in accordance with their resistance phenotypes. Moreover, though the one resistant *Salmonella* spp. was found with the *fosA* gene, it was sensitive to fosfomycin phenotypically. Similarly, a previous study showed that *Salmonella* spp. was carrying a quinolone resistance gene, but was sensitive to levofloxacin and ciprofloxacin, which might be due to the non-expression of this gene [30].

The present study showed the existence of *SHV*, *tetA*, *floR*, *aadA1*, *aac*(*6_*)*-lb*, and *fosA* genes in colistin-resistant *Klebsiella pneumoniae* isolates, with *SHV* gene predominance. Colistin-resistant *Klebsiella pneumoniae* isolates recovered from humans in India were also reported genotypically resistant to various antibiotics, including *TEM*, *SHV*, *aadA2*, *aac*(*6′*)*-lb-cr*, *tetD*, *fosA* [41].

Various antibiotic resistance genes in bacteria found in poultry and poultry meat are regarded as a significant risk to human health. Combining any of the two antibiotic resistant genes can make it possible for microbes to acquire many genes at once and quickly form high-risk pathogenic organisms. A previous study in Canada has documented the potential risk of dissemination of drug resistant bacteria to humans via the consumption of poultry [42]. A study in Ghana suggested that chicken bacterial isolates and human bacterial isolates had a high degree of similarity, signalling that resistant bacteria could be spreading between humans and animals [43]. To stop the spread of such resistant microorganisms via contaminated foods, rigorous hygiene precautions are therefore required [44]. It is important to educate the public about such novel risks and to instruct those working in the veterinary and animal agriculture sectors on how to use antibiotics responsibly.

## 4. Materials and Methods

### 4.1. Bacterial Strains and Study Design

A total of 54 colistin-resistant *Enterobacteriaceae*, including *E*. *coli* (n = 32), *Salmonella* spp. (n = 16) and *Klebsiella pneumoniae* (n = 6) were used in this study, which were recovered from chicken meat, cloacal swabs (CS) and litter samples from supermarkets and poultry farms, in Selangor, Malaysia, from 2019 to 2021. These bacteria were isolated and identified using traditional culture and biochemical tests [45,46,47], and confirmed with PCR using species-specific gene primers [48,49,50]. Colistin resistance was confirmed with broth microdilution (BMD) assay as recommended by EUCAST, and colistin minimum inhibitory concentrations (MIC) were recorded. According to EUCAST, *Enterobacteriaceae* having an MIC > 2 µg/mL against colistin were defined as colistin resistant. The genomic DNA of the colistin-resistant isolates was assessed with conventional PCR to detect colistin resistance (*mcr*) gene variants (*mcr-1* to *mcr-10*), in which *E*. *coli* ATCC25922 and *E*. *coli* NCTC 13846 were used as negative control and positive control, respectively. All the colistin-resistant isolates (Table 5) were subjected to antimicrobial susceptibility tests for various classes of antibiotics for the phenotypic profile and to determine the genotypic determinants of antibiotic resistance.

### 4.2. Phenotypic Antimicrobial Resistance Testing

The colistin-resistant *E*. *coli*, *Salmonella* spp., and *K*. *pneumoniae* isolates were phenotypically tested for their susceptibility to various classes of antibiotics, including aminoglycosides (S, Streptomycin; CN, Gentamycin; TOB, Tobramycin), fluoroquinolones (CIP, Ciprofloxacin; NOR, Norfloxacin; NA, Nalidixic acid), tetracyclines (TE, Tetracycline), cephalosporin (CTX, Cefotaxime; CRO, Ceftriaxone), fosfomycins (FOS, Fosfomycin) and chloramphenicol (C) by the Kirby–Bauer disk diffusion standard method [51,52]. A fresh culture of each isolate, with a concentration of 0.5 McFarland standard, was plated on Mueller–Hinton agar (Oxoid, UK) plates and incubated at 37 °C for 18–24 h. The results of the test were measured with digital slide callipers and recorded according to the Clinical and Laboratory Standards Institute (CLSI) guidelines [51].

#### 4.2.1. Rates of Antimicrobial Resistance (AMR)

The following calculation was used to determine the percentage of resistant isolates for each antibiotic.
$$\%\;\mathrm{rate}=\frac{\mathrm{Number}\;\mathrm{of}\;\mathrm{resistant}\;\mathrm{isolates}\;\times \;100}{\mathrm{Number}\;\mathrm{of}\;\mathrm{tested}\;\mathrm{isolates}}$$

According to Papadopoulos et al. (2021) and Adebowale et al. (2022), rates (%) of resistance were classified into % rate >70%, >50 to 70%, >20 to 50%, >10 to 20%, >1 to 10%, 0.1 to 1% and <0.1% as extremely high, very high, high, moderate, low, very low and rare, respectively [25,53].

#### 4.2.2. Multidrug Resistance (MDR) Patterns

Drug resistance was categorized based on Sweeney et al. [54]. Briefly, the MDR isolates were defined as the isolates that were non-susceptible to at least three antibiotics classes [54].

#### 4.2.3. The Multiple Antibiotic Resistance Index (MARI)

The formula: a/b, was used to calculate and interpret the multiple antibiotic resistance index (MARI), where “a” represents how many antibiotics an isolate proved resistance to, and “b” represents how many antibiotics were tested in total [36,55,56,57]. MARI values were compared to the threshold value of 0.2. A MARI value >0.2 implies that the original sources of isolates likely involved heavy antibiotic use or misuse of antibiotics [35], whereas <0.2 denotes samples from sources with less antibiotic usage or low risk [55]

### 4.3. Detection of Antimicrobial Resistance Genes

#### 4.3.1. DNA Extraction

The extraction of genomic DNA of all colistin-resistant isolates, including *E*. *coli*, *Salmonella* spp. and *K*. *pneumoniae* isolates, was performed by using the commercially available DNeasy^®^ Blood & Tissue Kit (QIAGEN GmbH, Hilden, Germany).

#### 4.3.2. Polymerase Chain Reaction

The genomic DNA of colistin-resistant isolates was subjected to PCR for the detection of antimicrobial-resistant genes against specific antibiotics, including β-lactamase genes (^bla^*CTX-M*, ^bla^*TEM*, and ^bla^*SHV*), Aminoglycosides (Gentamycin, *aac-3-IV*, Streptomycin, *aadA1*), Fluoroquinolones (Ciprofloxacin, *aac*(*6-*)*-lb*), Tetracyclines (Tetracycline, *tetA*), Chloramphenicol (*catA1* and *floR*) and Fosfomycins (Fosfomycin, *fosA*) by previously designed oligonucleotide primers and protocols (Appendix A) [36,58,59,60,61,62,63,64].

The PCR assay was carried out in 25 µL reaction mixture containing 2× master mix (MyTaq^TM^ Red Mix, Bioline, UK), each with forward and reverse primers (10 pmol/µL), PCR-grade water, and DNA template (Appendix A) using an Eppendorf Mastercycler pro S (Hamburg, Germany).

#### 4.3.3. Agarose Gel electrophoresis

The PCR products were electrophoresed at 80 V for 60 min in a gel electrophoresis system (Enduro, Labnet, Taiwan) through 1.5% (w/v) agarose gel (GeneDireX, New Jersey, NJ, USA) prepared in 0.5× TBE buffer containing 0.04 µL/mL nucleic acid staining (ETB “out” Nucleic Acid, Cat. No. FYD007-200P, Yestern Biotech Co. ltd, Taiwan). Aliquots of 5 µL of PCR products were applied to the gel. Depending on the sizes of amplicons, a 100 bp DNA ladder RTU (Cat. No. DM001-R500, GeneDireX) was used as a size marker in each gel. Using the Alpha Innotech gel documentation system (AlphaImager 2200, Haverhill, MA, USA), expected bands for the relevant genes (Appendix A) were seen and captured in photos under UV light. The obtained results of β-lactamase genes were verified by uniplex PCR analysis with the relevant gene.

### 4.4. Statistical Analysis

The differences in antibiotic resistance rates in colistin-resistant isolates (*Salmonella* spp., *E*. *coli* and *K*. *pneumoniae* strains) among sources (raw chicken meat, cloacal swab, litter) were tested using the chi-square (χ2) test in SPSS software v. 25.0 (IBM, Armonk, NY, USA). Statistical significance was considered at a *p*-value < 0.05.

## 5. Conclusions

Various numbers of antibiotic resistance genes were observed in the colistin-resistant *Enterobacteriaceae* isolates (*E*. *coli*, *Salmonella* spp., *K*. *pneumoniae*) tested in this current study. Most of the isolates were found to have multidrug resistance (MDR), with resistance to up to seven classes of antibiotics. The *mcr* gene of *Enterobacteriaceae* isolates was found to co-exist with multiple antibiotic resistance genes. Specifically, *^bla^TEM*, *tetA*, *floR*, *catA1*, *aac-3-IV*, *aadA1*, *fosA*, *aac*(*6_*)*-lb* genes co-existed with *mcr-1* in *E*. *coli* and *^bla^TEM*, *tetA*, *floR*, *aac-3-IV*, *fosA*, *aac*(*6_*)*-lb* genes co-existed with *mcr-5* in *Salmonella* spp. Colistin-resistant *K*. *pneumoniae* isolates were also detected with various numbers of resistance genes. The co-existence of several clinically important antimicrobial resistance genes in the present study exacerbates the antibiotic resistance problem in Malaysia and highlights significant concerns about potential future threats to infection treatment choices in humans and animals. This research will help to prepare the AMR surveillance and monitoring guidelines for policymakers throughout the country to mitigate the risk of AMR in humans and animals.

## Figures and Tables

**Figure 1 antibiotics-12-01060-f001:**
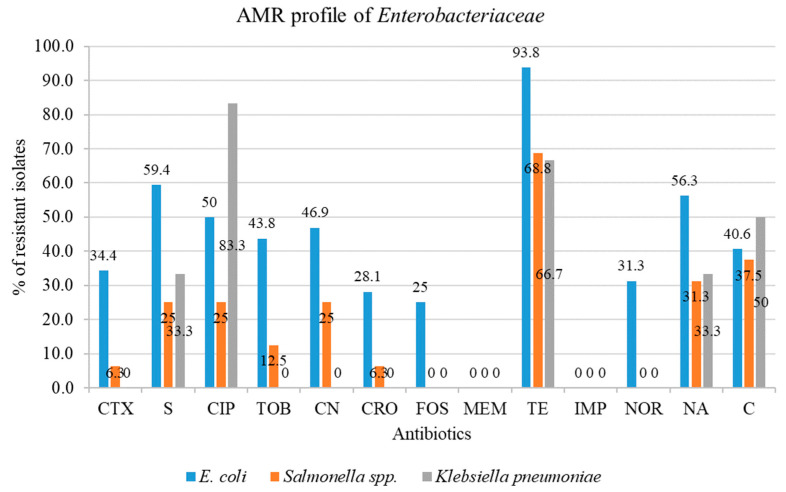
Species-wise representation of antibiotics resistance pattern of the *Enterobacteriaceae* isolates. CTX = Cefotaxime, S = Streptomycin, CIP = Ciprofloxacin, TOB = Tobramycin, CN = Gentamycin, CRO = Ceftriaxone, FOS = Fosfomycin, MEM = Meropenem, TE = Tetracycline, IMP = Imipenem, NOR = Norfloxacin, NA = Nalidixic acid, C = Chloramphenicol.

**Figure 2 antibiotics-12-01060-f002:**
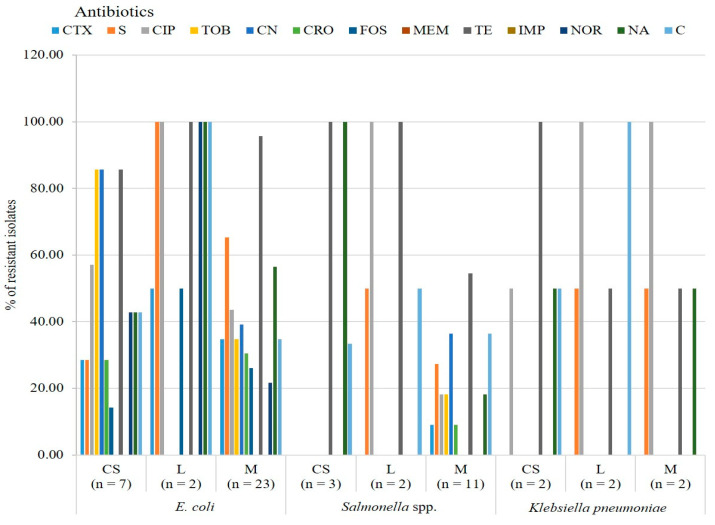
Antibiotic resistance patterns of *E*. *coli*, *Salmonella* spp. and *Klebsiella pneumoniae* isolated from different sources. CTX = Cefotaxime, S = Streptomycin, CIP = Ciprofloxacin, TOB = Tobramycin, CN = Gentamycin, CRO = Ceftriaxone, FOS = Fosfomycin, MEM = Meropenem, TE = Tetracycline, IMP = Imipenem, NOR = Norfloxacin, NA = Nalidixic acid, C = Chloramphenicol, CS = Cloacal swab, L = Litter, M = Meat.

**Figure 3 antibiotics-12-01060-f003:**
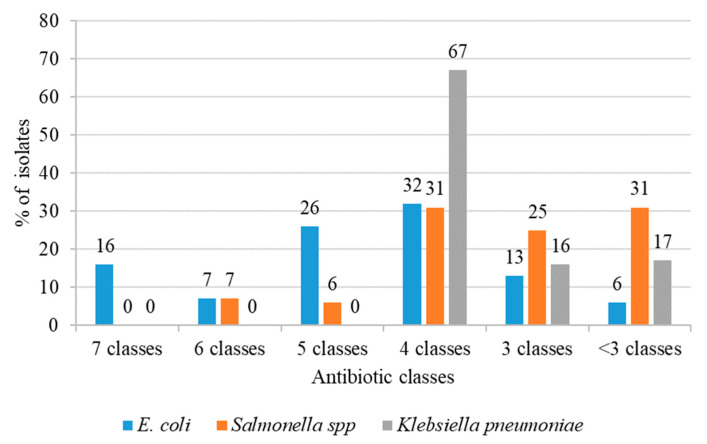
MDR patterns of colistin-resistant *Enterobacteriaceae* isolates.

**Figure 4 antibiotics-12-01060-f004:**
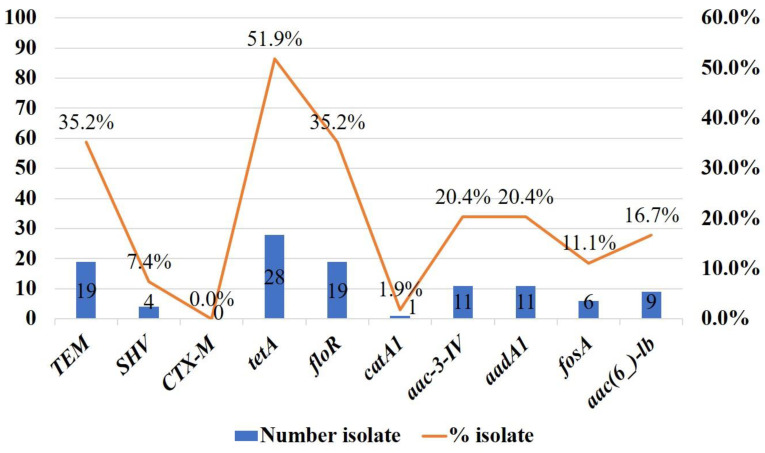
Co-existence of antimicrobial resistant genes in colistin-resistant *Enterobacteriaceae*.

**Figure 5 antibiotics-12-01060-f005:**
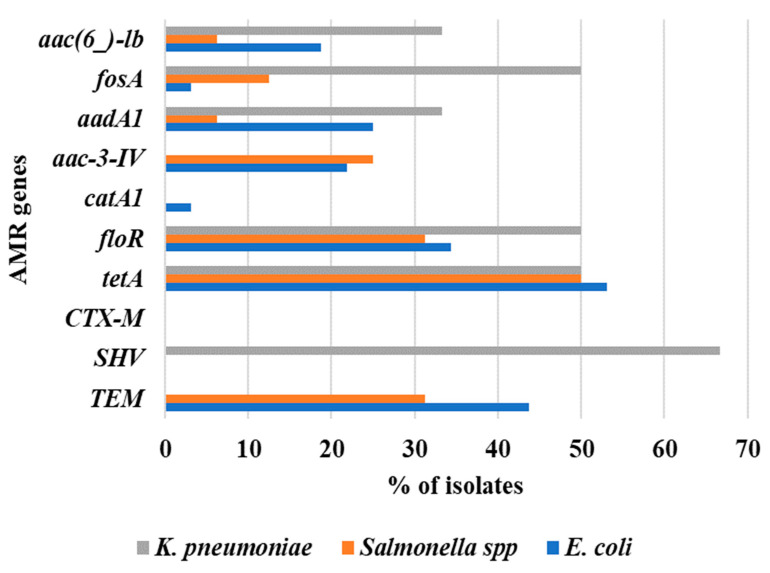
Species-wise co-existence of antimicrobial resistant genes in colistin-resistant *Enterobacteriaceae*.

**Table 1 antibiotics-12-01060-t001:** Multidrug resistance profile of *mcr*-harbouring *Enterobacteriaceae* isolates.

Isolates	No. of Antibiotics (Class)	Multidrug Profile	No. of Isolates (%)	Prevalence of MDR%
*mcr* carriage *Enterobacteriaceae* (n = 7)	1 (1)	Any one of the tested antibiotics	0	100
	2 (2)	Combination of any two antibiotics	0	
	5 (4)	TOB, CN, TE, C, CL	1 (14.28)	
	6 (5)	TOB, CN, NA, CIP, NOR, CL	1 (14.28)	
	7 (6)	S, TE, NA, CIP, NOR, C, CL	2 (28.57)	
	10 (6)	TOB, CN, S, TE, NA, CIP, CTX, CRO, C, CL		
	9 (7)	S, TE, NA, CIP, NOR, CTX, FOS, C, CL	3 (42.86)	
	11 (7)	TOB, CN, S, TE, NA, CIP, CTX, CRO, FOS, C, CL		
	11 (7)	TOB, CN, S, TE, CIP, NOR, CTX, CRO, FOS, C, CL		

Aminoglycosides (TOB = Tobramycin, CN = Gentamycin, S = Streptomycin), Fluoroquinolones (NA = Nalidixic acid, CIP = Ciprofloxacin, NOR = Norfloxacin), Tetracyclines (TE = Tetracycline), Cephalosporin (CTX = Cefotaxime, CRO = Ceftriaxone), Fosfomycins (FOS = Fosfomycin), Chloramphenicol (C), Polymyxins (CL = Colistin), MDR = Multidrug resistance.

**Table 2 antibiotics-12-01060-t002:** Prevalence of AMR genes in *Enterobacteriaceae* isolated from different sources.

Sources	Antibiotic Resistance Genes Profile (%)																			
	* ^bla^ * *TEM*	*p*-Value	* ^bla^ * *SHV*	*p*-Value	* ^bla^ * *CTX-M*	*p*-Value	*tetA*	*p*-Value	*floR*	*p*-Value	*catA1*	*p*-Value	*aac-3-IV*	*p*-Value	*aadA1*	*p*-Value	*fosA*	*p*-Value	*aac*(*6′*)*-Ib*	*p*-Value
Cloacal swab (n = 12)	5 (41.7)	0.868	1 (8.3)	0.03	0	nc	9 (75)	0.025	4 (33.3)	0.03	0	0.775	4 (33.3)	0.45	4 (33.3)	0.002	2 (16.7)	0.105	4 (33.3)	0.067
Litter (n = 6)	2 (33.3)		2 (33.3)		0		5 (83.3)		5 (83.3)		0		1 (16.7)		4 (66.7)		2 (33.3)		2 (33.3)	
Meat (n = 36)	12 (33.3)		1 (2.8)		0		14 (38.9)		10 (27.8)		1 (2.8)		11 (20.4)		3 (8.3)		2 (5.6)		3 (8.3)	
Total (n = 54)	19 (35.2)		4 (7.4)		0		28 (51.9)		19 (35.2)		1 (1.9)		11 (20.4)		6 (11.1)		9 (16.7)		9 (16.7)	

nc = not computed.

**Table 3 antibiotics-12-01060-t003:** Antimicrobial resistance gene (ARG) patterns in colistin-resistant *Enterobacteriaceae* (*E*. *coli*, n = 32; *Salmonella* spp., n = 16 and *K*. *pneumoniae*, n = 6).

Isolates	Strains	ARG Patterns	No. of ARG	No. of Isolates (%)
*E*. *coli*	E297	*^bla^TEM*, *tetA*, *floR*, *aac-3-IV*, *aadA1*, *fosA*, *aac*(*6_*)*-lb*	7	1 (3.13)
*E*. *coli*	E48	*^bla^TEM*, *tetA*, *floR*, *aac-3-IV*, *aadA1*, *aac*(*6_*)*-lb*	6	1 (3.13)
*E*. *coli*	E172	*tetA*, *floR*, *aac-3-IV*, *aadA1*, *aac*(*6_*)*-lb*	5	1 (3.13)
*E*. *coli*	E49	*^bla^TEM*, *tetA*, *floR*, *aadA1*, *aac*(*6_*)*-lb*	5	1 (3.13)
*E*. *coli*	E278	*^bla^TEM*, *floR*, *catA1*, *aac-3-IV*, *aac*(*6_*)*-lb*	5	1 (3.13)
*E*. *coli*	E275	*^bla^TEM*, *tetA*, *floR*, *aac-3-IV*	4	1 (3.13)
*E*. *coli*	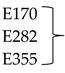			
*E*. *coli*	*^bla^TEM*, *tetA*, *floR*, *aadA1*	4	4 (12.5)
*E*. *coli*			
*E*. *coli*	E446			
*E*. *coli*	E331	*^bla^TEM*, *tetA*, *floR*, *aac*(*6_*)*-lb*	4	1 (3.13)
*E*. *coli*	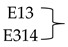	*^bla^TEM*, *tetA*, *aac-3-IV*	3	2 (6.25)
*E*. *coli*			
*E*. *coli*	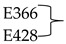	*^bla^TEM*, *tetA*	2	2 (6.25)
*E*. *coli*			
*E*. *coli*	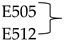	*tetA*	1	2 (6.25)
*E*. *coli*			
*E*. *coli*	-	-	0	15 (46.87)
*Salmonella* spp.	S283	*^bla^TEM*, *tetA*, *floR*, *aac-3-IV*, *fosA*, *aac*(*6_*)*-lb*	6	1 (6.25)
*Salmonella* spp.	S242	*^bla^TEM*, *tetA*, *floR*, *aadA1*, *fosA*	5	1 (6.25)
*Salmonella* spp.	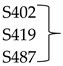			
*Salmonella* spp.	*^bla^TEM*, *tetA*, *floR*, *aac-3-IV*	4	3 (18.75)
*Salmonella* spp.			
*Salmonella* spp.	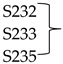			
*Salmonella* spp.	*tetA*	1	3 (18.75)
*Salmonella* spp.			
*Salmonella* spp.	-	*-*	0	8 (50)
*K*. *pneumoniae*	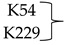	*^bla^SHV*, *tetA*, *floR*, *aadA1*, *fosA*, *aac*(*6_*)*-lb*	6	2 (33.3)
*K*. *pneumoniae*			
*K*. *pneumoniae*	K55	*^bla^SHV*, *tetA*, *floR*	3	1 (16.67)
*K*. *pneumoniae*	K402	*^bla^SHV*, *fosA*	2	1 (16.67)
*K*. *pneumoniae*	-	-	0	2 (33.3)

β-lactamase genes (*^bla^TEM* and *^bla^SHV*), Aminoglycosides (Gentamycin, *aac-3-IV*, Streptomycin, *aadA1*), Fluoroquinolones (Ciprofloxacin, *aac*(*6_*)*-lb*), Tetracyclines (Tetracycline, *tetA*), Chloramphenicol (*catA1* and *floR*) and Fosfomycins (Fosfomycin, *fosA*).

**Table 4 antibiotics-12-01060-t004:** Antimicrobial resistance gene (ARG) patterns in *mcr*-habouring colistin-resistant *Enterobacteriaceae* (*E*. *coli*, n = 6, *Salmonella* spp., n = 1).

Sources	Strains	Col-R Gene	Other ARG Patterns	No. of ARG	No. of Isolates (%)
Cloacal swab	E13	*mcr-1*	^bla^*TEM*, *tetA*, *aac-3-IV*	3	1 (14.3)
	E48	*mcr-1*	^bla^*TEM*, *tetA*, *floR*, *aac-3-IV*, *aadA1*, *aac*(*6_*)*-lb*	6	1 (14.3)
	E297	*mcr-1*	^bla^*TEM*, *tetA*, *floR*, *aac-3-IV*, *aadA1*, *fosA*, *aac*(*6_*)*-lb*	7	1 (14.3)
Litter	E172	*mcr-1*	*tetA*, *floR*, *aac-3-IV*, *aadA1*, *aac*(*6_*)*-lb*	5	1 (14.3)
Meat	E278	*mcr-1*	^bla^*TEM*, *floR*, *catA1*, *aac-3-IV*, *aac*(*6_*)*-lb*	5	1 (14.3)
	E331	*mcr-1*	^bla^*TEM*, *tetA*, *floR*, *aac*(*6_*)*-lb*	4	1 (14.3)
	S283	*mcr-5*	^bla^*TEM*, *tetA*, *floR*, *aac-3-IV*, *fosA*, *aac*(*6_*)*-lb*	6	1 (14.3)

β-lactamase genes (*^bla^TEM* and *^bla^SHV*), Aminoglycoside resistance genes (Gentamycin, *aac-3-IV*; Streptomycin, *aadA1*), Fluoroquinolone resistance genes (Ciprofloxacin, *aac*(*6-*)*-lb*), Tetracycline resistance genes (Tetracycline, *tetA*), Chloramphenicol resistance genes (*catA1* and *floR*) and Fosfomycin resistance genes (Fosfomycin, *fosA*), E = *E*. *coli*, S = *Salmonella* spp.

**Table 5 antibiotics-12-01060-t005:** List of colistin-resistant *Enterobacteriaceae* isolated from different samples of poultry and poultry meats.

Isolates	Sources	Number of Isolates
*E*. *coli* (n = 32)	Meat	23
	CS	7
	Litter	2
*Salmonella* spp. (n = 16)	Meat	11
	CS	3
	Litter	2
*K*. *pneumoniae* (n = 6)	Meat	2
	CS	2
	Litter	2
	Total	54

CS, cloacal swab.

## Data Availability

Not applicable.

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
