# Peer review of "Antimicrobial Resistance Profiles and Co-Existence of Multiple Antimicrobial Resistance Genes in mcr-Harbouring Colistin-Resistant Enterobacteriaceae Isolates Recovered from Poultry and Poultry Meats in Malaysia"

_antibiotics, 2023, doi:10.3390/antibiotics12061060_

Round 1

Reviewer 1 Report

Dear authors 

Please, find the comments below (Also, you can track the changes recommended and the comments within the attached version of the PDF file);

Comments to authors

antibiotics-2378717

Antimicrobial resistance profiles and co-existence of multiple antimicrobial resistance genes in mcr harbouring colistin-resistant Enterobacteriaceae isolates recovered from poultry and poultry meats in Malaysia

Item

Section

Page(s)

Line (s)

Comment(s)

1

Abstract

1

19

Add "d" to raise

2

Abstract

1

24

Please, use the full-name " Escherichia coli" and the form (E.coli) at first time

3

Abstract

1

25

Please, use the full-name " Klebsiella pneumoniae " and the form (K. pneumonia) at first time

4

Abstract

1

31

Please, add " In conclusion, the………………

5

Results

3

90

Please, delete the  extra "%" in  (31.3%%)

6

Results

5

127-129

Please, keep these abbreviations associated with the legends of Fig 3

7

Results

5

Fig. 4

Please, add the color keys of the isolates origin on the histogram , i.e CS (blue), L (orange) and M (grey)

8

Results

8

172- 184

Please replace "MAR index" with "MARI"

9

Results

9

188

Please replace "MAR index" with "MARI"

10

Results

10

194, 200

Please replace "MAR index" with "MARI"

11

Results

11

204, 207

Please replace "MAR index" with "MARI"

12

Results

12

211

Please replace "MAR index" with "MARI"

13

Discussion

20

298

For "MCR", did you mean MDR?

14

MM

23

435

Please, add "The Multiple Antibiotic Resistance index (MARI)" instead of  "The Multiple Antibiotic Resistance (MAR) index"

15

MM

23

437-439

Please replace "MAR index" with "MARI"

16

References

26-27

588-590

Please, check the reference No. (26)

17

References

27

591-592

Please, check the reference No. (27)

18

References

29

673-675

Please, add "doi: 10.3389/fmicb.2018.02299" to the reference No. (55)

Please, see the comments  within  the box above and within attached PDF file

Reviewer 2 Report

The study was to investigate the multiple antimicrobial resistance profiles and genes of the colistin-resistant Enterobacteriaceae isolated from poultry and poultry meats in Malaysia. The obtained data was valuable for understanding the occurrence of antimicrobial resistance in intestinal bacteria. However, the data presentation was bad. In this text, too many figures and tables was shown. It’s difficult to directly and easily get the points of the research points.

Line 25, the abbreviation of MDR should be explained.

Figure 1, 2, 3 and 4 should be combined into one figure. The Y axis in figure 2-4 showed “No. of resistent isolates”. In my opinion, using “persentage of resistent isolates is better understanding. Similarly, the figure 5-7 should be combined into one figure.

Line 151-154, it’s difficult for me to understand the explaination associated with Table 2.

Line 180, what’s MARI? Where’s the figure 8-10?!!!!!!

Table 3-5 should be combined or put them in supplimental material. Figure 11-13 were shown in supplimental material, followed by Table 7-9.

Reviewer 3 Report

This is an interesting study but it is not original because a large number of papers are present in literature on the same topic. However, the experiments are clear and well described. 

Minor comments:

  1. There are minor spelling and grammar mistakes
  2. All the figures especially fig 1 to 7 are not impressive, the data presentation could be improved and some figures can be combined together
  3. Please, check the English language

As the paper highlights the high incidence of the co-existence of several clinically

important antimicrobial resistance genes from poultry and poultry meat so indicate what future guidelines should be followed and what contributed to this in the local region.  

Minor revision 

Reviewer 4 Report

The manuscript “Antimicrobial resistance profiles and co-existence of multiple antimicrobial resistance genes in mcr harbouring colistin-resistant Enterobacteriaceae isolates recovered from poultry and poultry meats in Malaysia” describes the antibiotic resistance spectrum of 54 Enterobacteriaceae isolates resistant to colistin. Most of the isolates have been isolated from poultry meat (36) and only a small proportion from other sources (18). For each of the three representatives of the family Enterobacteriaceae, the sample is further reduced. Such a sample can only be described. It is impossible to obtain reliable statistical data on it.

When describing the results, first of all, it is necessary to show how the resistance of isolates to colistin was proved, and what genes were found in representatives of each genus.Then, leaving the minimum number of tables and figures, show to which spectrum of antibiotics the isolates were additionally resistant.

When mentioning a family, the name should be written in regular type, but not in italics.

The introduction does not provide information on the level of research into antibiotic resistance of Enterobacteriaceae in Malaysia.

Minor editing of English language required

Reviewer 5 Report

Carbapenem-resistant Enterobacteriaceae and multi-drug resistant Pseudomonas aeruginosa have low resistance rates to colistin, and therefore colistin antibiotics are back as one of the drugs of choice. The detection and characterisation of resistant strains of colistin in food and related environments is of public health importance. Md. Rezaul Karimd et al. isolated 54 strains of mcr-positive Enterobacteriaceae resistant to colistin from chicken and chicken meat, determined the resistance profile of colistin-resistant strains to antibiotics and the co-existence of resistance genes. 

The manuscript provides antibiotic phenotypic and genotypic data for mcr-positive Enterobacteriaceaes in local chicken-associated environments in some detail, but there are some issues that need to be revised to improve the quality of the manuscript.

1. Lines 45-47: Please check that the citation is appropriate. The original article appears to be about ESBL-producing bacteria. And please ensure that the original literature is cited. Please check to see if similar issues exist.

2. Figure 2 and Figure 3: The note about ‘CS\L\M’ is missing.

3. Whether similar figures and tables can be combined together, such as Figures 2, 3, and 4. The manuscript has too many diagrams and low information density.

4.  Line 232: Is '%' missing?

5.Lines 397-402: Please provide detailed sample information, such as: sample quantity, sampling time, etc. And please describe the steps of isolating and screening colistin-resistant bacteria in more detail.

6. The entire Discussion section should be condensed.

7. The manuscript can also be improved from the following aspects:

â…  WGS is currently a good method for studying bacterial epidemiology. This manuscript would be well enhanced  with WGS analysis.

â…¡ If WGS is not available, PFGE typing should also be done for the isolate, or ST types should be determined by PCR. A clear prevalence profile of the isolates could improve our understanding of colistin-resistant organisms.

â…¢ It should be supplemented whether the mcr gene is transferable.

Round 2

Reviewer 2 Report

I recommend to accept this paper.

Reviewer 5 Report

No further comments.